# The impact of COVID-19 workload on psychological distress amongst Canadian intensive care unit healthcare workers during the 1st wave of the COVID-19 pandemic: A longitudinal cohort study

Daniel Pestana[1,2☯], Kyra Moura[3☯], Claire Moura[3☯], Taylor Mouliakis[4], Frédérick D'Aragon[5,6], Jennifer L. Y. Tsang[7,8]*, Alexandra Binnie[1]

1 William Osler Health System, Department of Critical Care, Etobicoke, Ontario, Canada, 2 Algarve Biomedical Centre Research Institute, Faro, Portugal, 3 Department of Anesthesiology, Pharmacology and Therapeutics, University of British Columbia, Vancouver, British Columbia, Canada, 4 School of Medicine, Queen's University, Kingston, Ontario, Canada, 5 Department of Anesthesiology, Université de Sherbrooke, Sherbrooke, Quebec, Canada, 6 Centre de Recherche du Centre Hospitalier Universitaire de Sherbrooke, Sherbrooke, Quebec, Canada, 7 Niagara Health, St. Catharines, Ontario, Canada, 8 Department of Medicine, McMaster University, Hamilton, Ontario, Canada

☯ These authors contributed equally to this work.
* jennifer.tsang@mcmaster.ca

## Abstract

Intensive care unit healthcare workers (ICU HCW) are at risk of mental health disorders during emerging disease outbreaks. Numerous cross-sectional studies have reported psychological distress, anxiety, and depression amongst ICU HCW during the COVID-19 pandemic. However, few studies have followed HCW longitudinally, and none of these have examined the association between COVID-19 workload and mental health. We conducted a longitudinal cohort study of 309 Canadian ICU HCW from April 2020 to August 2020, during the 1st wave of the COVID-19 pandemic. Psychological distress was assessed using the General Health Questionnaire 12-item scale (GHQ-12) at 3 timepoints: during the acceleration phase of the 1st wave (T1), the deceleration phase of the 1st wave (T2), and after the 1st wave had passed (T3). Clinically relevant psychological distress, defined as a GHQ-12 score ≥ 3, was identified in 64.7% of participants at T1, 41.0% at T2, and 34.6% at T3. Psychological distress was not associated with COVID-19 workload at T1. At T2, psychological distress was associated with the number of COVID-19 patients in the ICU (odds ratio [OR]: 1.06, 95% confidence interval [CI]: 1.00, 1.13) while at T3, when COVID-19 patient numbers were low, it was associated with the number of weekly hospital shifts with COVID-19 exposure (OR: 1.33, 95% CI: 1.09, 1.64). When analyzed longitudinally in a mixed effects model, pandemic timepoint was a stronger predictor of psychological distress (OR: 0.24, 95% CI: 0.15, 0.40 for T2 and OR: 0.16, 95% CI: 0.09, 0.27 for T3) than COVID-19 workload. Participants who showed persistent psychological distress at T3 were compared with those who showed recovery at T3. Persistent psychological distress was associated with a higher number of weekly shifts with COVID-19 exposure (OR: 1.97, 95% CI:1.33, 3.09) but not with a

**Data Availability Statement:** All relevant non-identifying data are within the manuscript and its Supporting information files.

**Funding:** This study was funded by an unrestricted research grant to JT from Mohawk Medbuy, a not-for-profit, shared services corporation (https://www.mohawkmedbuy.ca/). The funder had no role in study design, data collection and analysis, decision to publish, or preparation of the manuscript.

**Competing interests:** The authors have declared that no competing interests exist.

higher number of COVID-19 patients in the ICU (OR: 0.86, 95% CI: 0.76, 0.95). In summary, clinically relevant psychological distress was observed in a majority of ICU HCW during the acceleration phase of the 1st wave of the COVID-19 pandemic but decreased rapidly as the 1st wave progressed. Persistent psychological distress was associated with working more weekly shifts with COVID-19 exposure but not with higher numbers of COVID-19 patients in the ICU. In future emerging disease outbreaks, minimizing shifts with direct disease exposure may help alleviate symptoms for individuals with persistent psychological distress.

## Introduction

Mental health issues are common amongst healthcare workers (HCW) during emerging disease outbreaks [1–4]. Risk factors include increased workload, shortages of personal protective equipment (PPE), anxiety with respect to being infected, anxiety with respect to transmitting the infection to family and friends, and loss of social supports due to self-isolation and quarantine [5, 6]. During the 1st wave of the COVID-19 pandemic, cross-sectional studies in multiple countries reported high rates of psychological morbidity amongst HCW, including psychological distress, anxiety, depression, and post-traumatic stress disorder (PTSD). A meta-analysis of 13 studies reported a pooled prevalence of 23.2% for symptoms of anxiety and 22.8% for symptoms of depression [7].

Intensive care unit (ICU) workers may be at higher risk of mental health issues during emerging disease outbreaks than other HCW. At the peak of the 1st wave of the pandemic, a study of 1,058 French ICU HCW reported anxiety, depression and peritraumatic dissociation in 50.4%, 30.4%, and 32% of participants, respectively [8]. A separate study of 2,643 French ICU HCW reported clinically relevant psychological distress in 55% of respondents [9]. The prevalence was higher in "high intensity" areas of the country, where the number of COVID-19 ICU patients exceeded the number of baseline ICU beds. In a study of 709 ICU HCW in the UK, 45% met criteria for depression, PTSD, severe anxiety, or problem drinking and 13% reported frequent thoughts of suicidality or self-harm [10]. In all three studies, mental health issues were more common amongst females than males, and more common amongst nurses and nursing assistants than physicians.

Studies conducted after the peak of the 1st wave have reported lower rates of psychological distress. A Canadian study of ICU HCW, conducted between June and August of 2020, reported psychological distress in only 18% of respondents [11]. However, few studies have followed ICU HCW longitudinally to examine changes in mental health over time [12, 13]. None of these, to our knowledge, have examined the relationship between COVID-19 workload and mental health.

In this longitudinal study, we followed Canadian ICU HCW from April to July 2020, during the 1st wave of the pandemic, to identify variables associated with psychological distress as well as variables associated with persistence of psychological distress over time.

## Materials and methods

### Survey design

The study was designed as a 4-month, longitudinal survey of Canadian ICU HCW. It was built in an expedited fashion to capture data during the 1st wave of the COVID-19 pandemic. At study entry (T1) in April 2020, participants provided baseline demographic data including sex,

age, profession, and years of experience. Psychological well-being was assessed using the General Health Questionnaire 12-item scale (GHQ-12), a self-reported screening tool that focuses on breaks in normal functioning rather than life-long traits [14–16].

After enrolment, participants were asked to complete weekly surveys reporting their COVID-19 workload in the previous 7 days, including the number of hospital shifts worked, the number of hospital shifts worked with exposure to COVID-19 patients, and the maximum number of COVID-19 patients in the ICU in the previous week.

The GHQ-12 was repeated at two timepoints during the 16 week study: during the deceleration phase of the 1st wave (T2), between June 1st and June 7th, 2020, and after the 1st wave of the pandemic had passed (T3), between July 31st and August 14th, 2020.

## Survey administration

The survey was administered online through the secure Qualtrics platform (www.qualtrics.com). The target population was Canadian ICU HCW, including physicians, registered nurses, respiratory therapists, and allied health professionals (i.e. physiotherapists, speech and language therapists, dieticians, pharmacists, clerks and healthcare assistants). The survey link was disseminated through online advertisements (S1 Appendix) to the membership of the Canadian Critical Care Trials Group (CCCTG), the Canadian Community ICU Research Network (CCIRNet), the Canadian Critical Care Society (CCCS), and the Critical Care Nurses Association, as well as through their associated social media groups. Distribution began on April 6th, 2020 and enrollment closed on April 30th, 2020 [17]. Anyone with access to the link could potentially enrol. Due to the broad distribution strategy, it was not possible to calculate a response rate.

Electronic consent (e-consent) was provided by participants prior to completion of the T1 survey. Withdrawal was allowed at any time and participants could request removal of their data. Participants who did not reply to two consecutive surveys were sent an email asking if they wished to continue in the study. During the 16 weeks of the study, 64 participants withdrew from further participation and 1 participant asked for their data to be removed.

To ensure data linkage, weekly surveys were distributed using unique email survey links. Each survey remained accessible for 7 days and a reminder email was sent 3 days after distribution. The T2 and T3 surveys were distributed in conjunction with the 9th and 16th weekly surveys on June 1st, 2020 and July 31st, 2020, respectively. Responses to the final survey closed on August 14th, 2020.

## Data analysis

Data were described as number and percentage for categorical variables and mean and standard deviation or median and interquartile range for continuous variables. Response categories on the GHQ-12 questionnaire were coded using the 0-0-1-1 scoring method [14]. A threshold of $\geq 3$ points on the GHQ-12 has been identified as an appropriate cutoff for mental health diagnosis screening [15, 16] and a marker of clinically relevant psychological distress [2, 18].

Multivariable logistic regression was used to identify predictors of clinically relevant psychological distress (GHQ-12 $\geq 3$). Variables included age, sex, profession, years of experience, and COVID-19 workload variables. Stepwise forwards and backwards regression was used to identify the model with the lowest Akaike Information Criterion (AIC) [19]. Sex and profession were included in all models because previous studies have shown a strong association between these variables and HCW mental health during emerging disease outbreaks [8–10]. For analysis of longitudinal data, a mixed effects model was created in which participant

identity was included as a random effect, sex and profession as fixed effects, and age, years of experience, COVID-19 workload variables and pandemic timepoint as optional fixed effects. To deal with missingness, median weekly hospital shifts, median weekly hospital shifts with COVID-19 exposure, and the median number of patients in the ICU were used as variables. Stepwise forwards and backwards regression was used to identify the model with the lowest AIC. The minimal dataset is available as Supporting Information (S1 Dataset).

All analysis was performed using R software v 3.6.2 [20].

### Ethical consideration

Ethics approval was obtained from the Hamilton Integrated Research Ethics Board (HiREB #10790) and the Centre Hospitalier Universitaire de Sherbrooke (#MP-31-2021-3704).

## Results

Study enrolment began on April 6th, 2020, during the acceleration phase of the 1st wave of the pandemic in Canada. A total of 309 Canadian ICU healthcare workers (HCW) enrolled in the study by completing the e-consent and T1 survey. Although proof of institutional affiliation was not required, 129 respondents (42%) provided email addresses from Canadian hospitals, medical systems, or universities, representing 33 different institutions. A majority of participants were working in the province of Ontario (78%), with the remainder in Quebec, Alberta, Manitoba, the Atlantic provinces and British Columbia. Registered nurse (RN) was the most common profession (47.9%), followed by physician (26.5%), respiratory therapist (RT) (14.2%), and allied health professional (11.3%) (Table 1). The mean age of participants was 39.8 years (SD = 9.67) and a majority were female (73.5%).

After enrolment, participants were asked to complete weekly surveys tracking their COVID-19 workload. The average number of surveys completed was 9.23 (SD = 5.22). The GHQ-12 survey was repeated at week 9 (T2), with 195/309 (63.1%) respondents, and at week 16 (T3) with 191/309 (61.8%) respondents. Overall, 217/309 (70.2%) participants responded to either the T2 or T3 survey and 169/309 completed both. Participant characteristics were similar at T1, T2 and T3 (Table 1).

**Table 1. Participant demographics at T1, T2 and T3.**

| Timepoint | T1 N = 309 | T2 N = 195 | T3 N = 191 | p-value |
|---|---|---|---|---|
| **Age** | | | | |
| Mean (SD) | 39.8 (9.67) | 41.0 (10.1) | 41.4 (9.93) | 0.165 |
| Median [Min, Max] | 39.0 [23.0, 63.0] | 41.0 [23.0, 63.0] | 41.0 [23.0, 63.0] | |
| **Sex** | | | | |
| Female | 227 (73.5%) | 149 (76.4%) | 144 (75.4%) | 0.742 |
| Male | 82 (26.5%) | 46 (23.6%) | 47 (24.6%) | |
| **Role** | | | | |
| Allied Health | 35 (11.3%) | 24 (12.3%) | 25 (13.1%) | 0.941 |
| Physician | 82 (26.5%) | 52 (26.7%) | 55 (28.8%) | |
| Registered nurse | 148 (47.9%) | 97 (49.7%) | 87 (45.5%) | |
| Respiratory therapist | 44 (14.2%) | 22 (11.3%) | 24 (12.6%) | |
| **Years of Experience** | | | | |
| Mean (SD) | 13.8 (9.38) | 15.3 (9.73) | 15.4 (9.70) | 0.115 |
| Median [Min, Max] | 12.0 [1.00, 43.0] | 13.0 [1.00, 40.0] | 14.0 [1.00, 40.0] | |

## COVID-19 workload

COVID-19 workload was tracked weekly during the 16 weeks of the study. Variables included the number of weekly hospital shifts, the number of weekly shifts with direct exposure to COVID-19 patients, and the maximum number of COVID-19 patients in the participant's ICU. Fig 1 shows the change in COVID-19 workload variables over the 16 weeks of the study. Weekly hospital shifts were relatively constant (Fig 1A), while weekly hospital shifts with exposure to COVID-19 patients (Fig 1B) and the maximum number of COVID-19 patients in the ICU (Fig 1C) both declined, in keeping with the progression of the 1st wave of the pandemic [17].

## Prevalence of clinically relevant psychological distress

At T1, during the acceleration phase of the 1st wave of the pandemic, 200 of 309 study participants (64.7%) scored $\geq$ 3 points on the General Health Questionnaire 12-item scale (GHQ-12), indicating clinically relevant psychological distress (Table 2). At T2, during the deceleration phase, 80/195 participants (41.0%) scored $\geq$ 3 points on the GHQ-12. At T3, after the 1st wave had passed, 66/191 participants (34.6%) scored $\geq$ 3 points on the GHQ-12. Similar results were observed in the subgroup of 169 participants (54.7%) who completed all three GHQ-12 surveys ("complete participants") (Table 2).

## Longitudinal change in GHQ-12 scores

Average GHQ-12 scores declined from T1 to T3 (Anova, p < 0.0001) (Fig 2A). At T1 the average GHQ-12 score was 3.85 (SD = 2.78), at T2 it was 2.65 (SD = 2.83), and at T3 it was 2.28 (SD = 3.01). A similar decline was seen amongst the 169 participants who completed the T1, T2 and T3 surveys (Anova, p < 0.0001) (Fig 2B).

Female participants reported higher GHQ-12 scores than male participants at all timepoints (Fig 2C). However, both sexes showed a significant decline in GHQ-12 scores from T1 to T3 (Fig 2C). Physicians, registered nurses, and respiratory therapists also showed a decrease in GHQ-12 scores between T1 and T3 (Fig 2C). Allied health professionals did not show a significant change in GHQ-12 scores; however, the number of participants in this category was small and it represented a diverse group of professionals (Fig 2D).

## Variables predicting psychological distress

Multivariable logistic regression with stepwise variable selection was used to identify variables that predicted clinically relevant psychological distress (GHQ-12 score $\geq$ 3) at each timepoint (Table 3). Sex and profession were included in all models due to their known association with HCW mental health during emerging disease outbreaks [8–10, 21].

At T1, during the acceleration phase of the 1st wave, the best regression model included only sex and profession, without any of the COVID-19 workload variables (Table 3). Female participants were at higher risk of psychological distress than male participants, and nurses were at higher risk than other professionals. At T2, during the deceleration phase of the 1st wave, the best regression model included sex, profession, weekly hospital shifts, and the maximum number of COVID-19 patients in the participant's ICU. Notably, more weekly hospital shifts was associated with decreased psychological distress (odds ratio (OR): 0.76, 95% confidence interval (CI): 0.59, 0.98). At T3, the best regression model included sex, profession, and weekly hospital shifts with direct exposure to COVID-19 patients (OR 1.33, 95% CI: 1.09, 1.64) (Table 3).

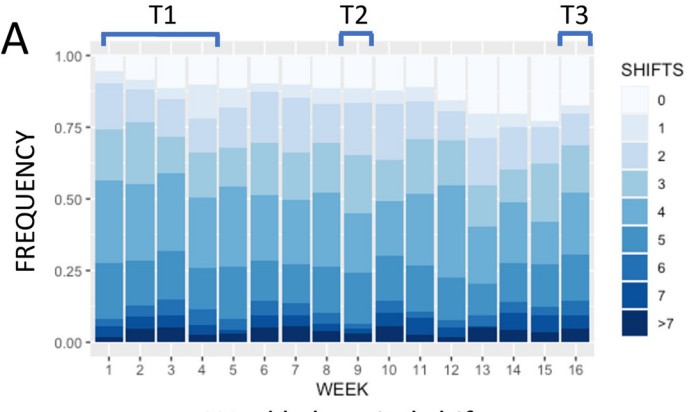

Weekly hospital shifts

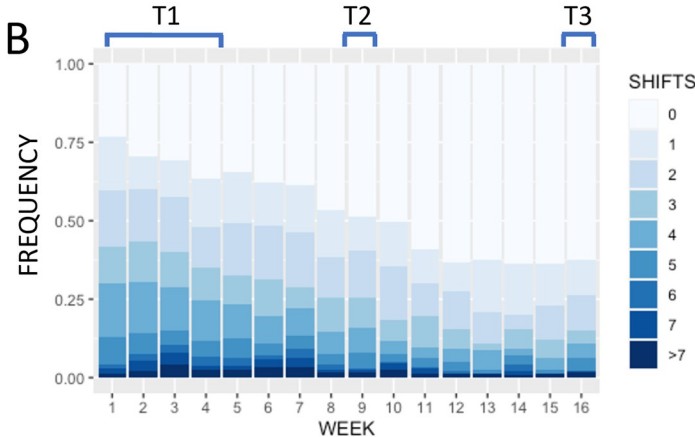

Weekly hospital shifts with
COVID-19 exposure

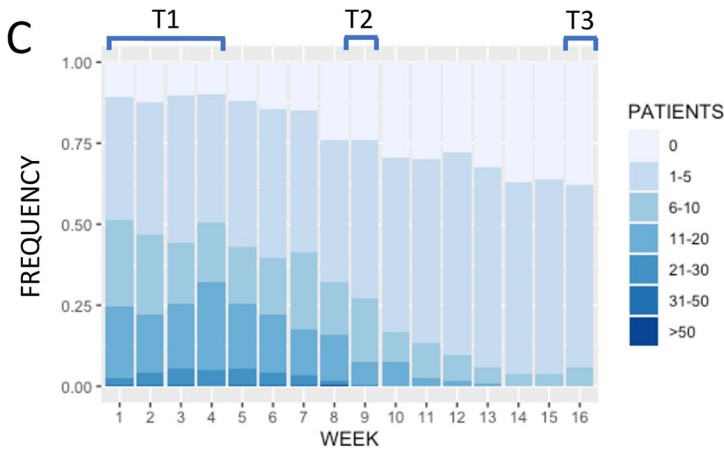

# of COVID-19 patients in ICU

**Fig 1. Weekly COVID-19 workload data reported by participants during the 16 weeks of the study.** (A) Weekly hospital shifts. (B) Weekly hospital shifts with exposure to COVID-19 patients. (C) The maximum number of COVID-19 patients admitted to the participant's ICU. Study weeks are shown on the x axis and the frequency of each response is shown on the y axis. T1, T2 and T3 indicate the timing of the GHQ-12 surveys.

**Table 2. Prevalence of clinically relevant psychological distress (GHQ-12 score ≥ 3) at T1, T2 and T3 amongst (A) all participants and (B) participants who completed all GHQ-12 surveys surveys (n = 169).**

|   | Timepoint | T1 | T2 | T3 |
|---|-----------|----|----|----|
| A | All participants | 200/309 (64.7%) | 80/195 (41.0%) | 66/191 (34.6%) |
| B | Complete participants | 105/169 (62.1%) | 68/169 (40.2%) | 53/169 (31.4%) |
|   | *Chi-Square P value* | 0.642 | 0.963 | 0.596 |

A mixed effects model was created to identify variables predictive of clinically relevant psychological distress over time (Table 3). In addition to sex and profession, age, years of experience, COVID-19 workload variables and pandemic timepoint (T1 vs T2 vs T3) were analysed as potential predictors. Given the longitudinal nature of the data, participant identity was included as a random effect. The best regression model included sex, profession, and pandemic timepoint, but not individual COVID-19 workload variables (Table 3).

## Persistent psychological distress

Out of the 190 participants who reported psychological distress at T1, 122 responded to the T3 survey. Of these, 55/122 (45.1%) reported a GHQ-12 score ≥ 3 at T3, suggesting persistent psychological distress, whilst 67/122 (54.9%) reported a GHQ-12 score < 3. No statistically significant differences in demographics or COVID-19 workload variables were identified between individuals with persistent psychological distress vs those with recovery (Table 4). The Brief Resilience Coping Scale (BRCS) [22], a self-reported measure of resilience, was also applied at T3 and no difference in resilience was identified between the two groups.

## Predictors of persistent psychological distress

Multivariable logistic regression was used to determine the association of age, sex, profession, years of experience and COVID-19 workload variables with persistent psychological distress. Workload variables were derived from all weekly survey responses and included the *median* number of weekly hospital shifts worked, the *median* number of weekly hospital shifts worked with exposure to COVID-19 patients, and the *median* number of COVID-19 patients in the participant's ICU.

The best regression model is shown in Table 5. The model included sex, profession, median weekly hospital shifts with exposure to COVID-19 and median number of COVID-19 patients in the participant's ICU. Weekly hospital shifts with exposure to COVID-19 patients was associated with persistent psychological distress. Conversely, male sex, RT or allied health profession, and the number of COVID-19 patients in the ICU were associated with recovery. The area under the receiver operating curve for the model was 0.74.

## Discussion

Emerging disease outbreaks place a heavy toll on HCW. During the COVID-19 pandemic, multiple studies have documented psychological morbidity amongst frontline HCW, particularly amongst ICU HCW [8, 12, 13, 21, 23, 24]. However, few studies have monitored the mental health of HCW over time. In this study, psychological distress was assessed in a cohort of Canadian ICU HCW at three timepoints during the 1st wave of the COVID-19 pandemic: the acceleration phase in April 2020, the deceleration phase in June 2020, and after the 1st wave had passed in July-August 2020 [14]. At each timepoint, participants completed the General Health Questionnaire 12-item scale (GHQ-12), which has been widely used to assess HCW

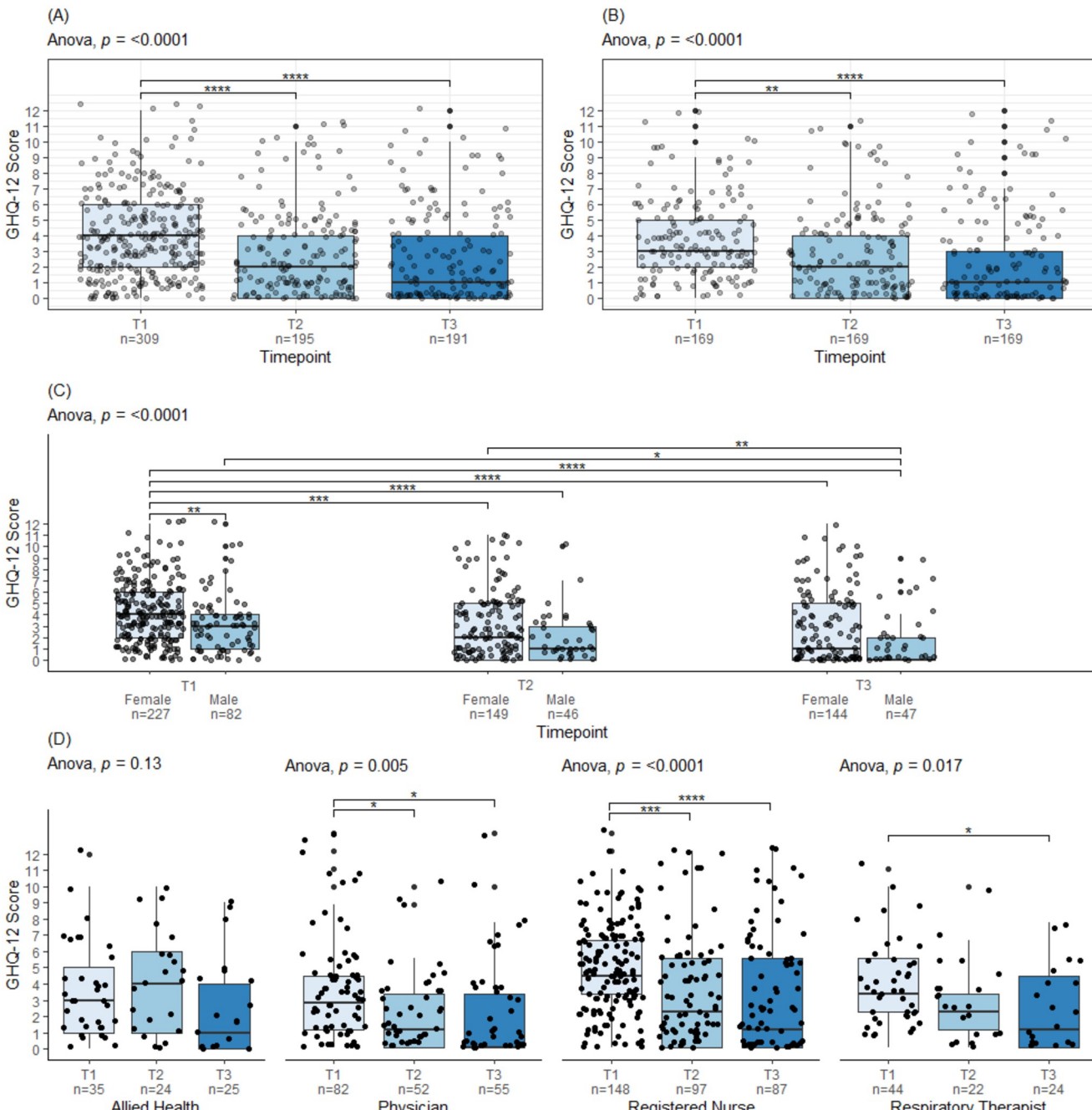

**Fig 2. GHQ-12 scores at T1, T2 and T3 for (A) all participants, (B) complete participants (n = 169), (C) all participants subdivided by profession, (D) all participants subdivided by sex.** Post-hoc Tukey test p-values are displayed as p≤0.05 (*), p≤0.01 (**), p≤0.001 (***), p≤0.0001 (****).

mental health during emerging disease outbreaks and is designed to assess breaks in normal functioning rather than life-long traits [14]. Although it does not diagnose specific psychiatric disease, the GHQ-12 has good specificity (74–88%) and sensitivity (74–80%) for mental health disorders, particularly anxiety and mood disorders [16, 25].

In April 2020, during the acceleration phase of the 1st wave in Canada, over 60% of participants showed evidence of clinically relevant psychological distress. At this early stage of the

**Table 3. Multivariate logistic regression models to predict clinically relevant psychological distress (GHQ-12 ≥ 3) at T1, T2, T3 and all timepoints.** Models were selected through stepwise forwards and backwards regression to obtain the lowest Akaike Information Criterion (AIC). In the final mixed effects model, participant ID was included as a random effect. OR: odds ratio. 95% CI: 95% confidence interval. AIC: Akaike Information Criterion. ROC: receiver operating curve. ICC: intraclass correlation coefficient. For weekly hospital shifts, the OR is the increase in risk per additional shift worked.

| Variables | T1 OR (95% CI) | p-value | T2 OR (95% CI) | p-value | T3 OR (95% CI) | p-value | All timepoints OR (95% CI) | p-value |
|---|---|---|---|---|---|---|---|---|
| **Sex: Male** | 0.66 (0.36, 1.20) | 0.2 | 0.59 (0.24, 1.43) | 0.3 | 0.36 (0.14, 0.89) | 0.033 | 0.43 (0.22, 0.85) | 0.016 |
| **Profession: Physician** | — | — | — | — | — | — | — | — |
| **Profession: RN** | 2.54 (1.34, 4.84) | 0.005 | 1.21 (0.45, 3.30) | 0.7 | 1.26 (0.54, 2.98) | 0.6 | 1.69 (1.11, 2.58) | 0.014 |
| **Profession: RT** | 1.45 (0.68, 3.14) | 0.3 | 1.19 (0.35, 4.04) | 0.8 | 0.82 (0.25, 2.48) | 0.7 | 1.56 (0.65, 3.73) | 0.32 |
| **Profession: Allied health** | 1.11 (0.48, 2.60) | 0.8 | 1.92 (0.59, 6.36) | 0.3 | 0.91 (0.29, 2.76) | 0.9 | 1.54 (0.60, 3.92) | 0.37 |
| **Median number of weekly hospital shifts** | | | 0.76 (0.59, 0.98) | 0.037 | | | | |
| **Median number of weekly hospital shifts with COVID-19 exposure** | | | | | 1.33 (1.09, 1.64) | 0.006 | | |
| **Median number of COVID-19 patients in the ICU** | | | 1.06 (1.00, 1.13) | 0.070 | | | | |
| **Timepoint: T2** | | | | | | | 0.24 (0.15, 0.40) | <0.001 |
| **Timepoint: T3** | | | | | | | 0.16 (0.09, 0.27) | <0.001 |
| **No. of observations** | 309 | | 166 | | 190 | | 695 | |
| **AIC** | 392.5 | | 230.0 | | 239.7 | | 862.7 | |
| **Area under the ROC** | 0.64 | | 0.64 | | 0.68 | | 0.69 | |
| **ICC** | | | | | | | 0.40 | |

**Table 4. A comparison of participants with "persistent" psychological distress at T3 (GHQ-12 ≥ 3 at T1 and T3) versus those without (GHQ-12 score ≥ 3 at T1 and < 3 at T3).** Q-value: p-value adjusted to maintain false discovery rate ≤ 0.05. Variables in bold are statistically significant at the p = 0.05 level. BRCS: Brief Resilience Coping Scale.

| Characteristic | Overall N = 122 | Persistence N = 55 | Recovery N = 67 | p-value | q-value |
|---|---|---|---|---|---|
| **GHQ-12 score at T1** | 5.2 (2.2) | 5.7 (2.5) | 4.8 (1.9) | 0.029 | 0.2 |
| **GHQ-12 score at T3** | **2.9 (3.2)** | **5.9 (2.4)** | **0.5 (0.7)** | **<0.001** | **<0.001** |
| **Age, mean (SD)** | 41.0 (9.9) | 41 (11) | 40.9 (8) | >0.9 | >0.9 |
| **Sex (% female)** | | | | 0.08 | 0.3 |
| Female | 98 (80%) | 48 (87%) | 50 (75%) | | |
| Male | 24 (20%) | 7 (13%) | 17 (25%) | | |
| **Profession** | | | | 0.2 | 0.4 |
| Allied Health | 12 (9.8%) | 3 (5.5%) | 9 (13%) | | |
| Physician | 29 (24%) | 14 (25%) | 15 (22%) | | |
| Registered Nurse | 65 (53%) | 33 (60%) | 32 (48%) | | |
| Respiratory therapist | 16 (13%) | 4 (9.1%) | 11 (16%) | | |
| **Resilience (BRCS)** | 15.1 (2.3) | 15.0 (2.1) | 15.2 (2.5) | 0.6 | 0.8 |
| **Median weekly hospital shifts** | 3.3 (1.4) | 3.5 (1.4) | 3.1 (1.5) | 0.13 | 0.4 |
| **Median weekly hospital shifts with COVID-19 exposure** | 1.2 (1.4) | 1.4 (1.5) | 1.0 (1.3) | 0.055 | 0.3 |
| **Median maximum COVID-19 patients in ICU** | 6.4 (5.5) | 5.72 (4.92) | 7.04 (5.91) | 0.2 | 0.4 |
| **Number of weekly responses** | 12.7 (3.0) | 12.2 (3.2) | 13.1 (2.8) | 0.12 | 0.3 |

**Table 5. Multivariable logistic regression to predict persistence of psychological distress at T3.**

| Variable | OR | 95% CI | p-value |
|---|---|---|---|
| **Profession: Physician** | — | — | — |
| **Profession: RN** | 0.60 | 0.20, 1.76 | 0.36 |
| **Profession: RT** | 0.16 | 0.03, 0.71 | 0.02 |
| **Profession: Allied health** | 0.17 | 0.03, 0.092 | 0.05 |
| **Sex: Male** | 0.27 | 0.08, 0.88 | 0.037 |
| **Weekly hospital shifts with exposure to COVID-19 patients** | 1.97 | 1.33, 3.09 | 0.002 |
| **Number of COVID-19 patients in the ICU** | 0.86 | 0.76, 0.95 | 0.005 |
| **Number of observations** | 121 | | |
| AIC | 158.4 | | |
| AUROC | 0.74 | | |

OR: odds ratio. 95% CI: 95% confidence interval. AIC: Akaike Information Criterion. AUROC: Area Under the Receiver Operating Curve.

pandemic, sex and profession were the best predictors of psychological distress while individual COVID-19 workload variables were not significant predictors. The high level of psychological distress observed at the beginning of the 1st wave is consistent with other studies of ICU HCW at similar timepoints [8, 9, 24]. April 2020 was a time of great anxiety and uncertainty worldwide; COVID-19 coverage accounted for more than 50% of headlines in major online news sites [26] and a significant increase in psychological distress was observed in the general population at that time [27]. Moreover, news stories were reporting many infections and fatalities amongst HCW in China and Italy during their devastating 1st waves [28, 29]. External factors, such as news exposure and uncertainty about the future course of the pandemic, may have had more impact on HCW mental health at the beginning of the 1st wave than did individual COVID-19 workload.

In June 2020, during the deceleration phase of the 1st wave, the prevalence of clinically relevant psychological distress decreased from 64.7% to 41.0% of participants. In addition to female sex and RN profession, the number of COVID-19 patients in the participant's ICU was predictive of psychological distress. Unexpectedly, the number of weekly hospital shifts worked (with or without COVID-19 exposure) was protective against psychological distress. It is unclear, however, whether working more shifts had a positive impact on mental health or whether HCW with better mental health were willing to take on more shifts.

In July 2020, after the 1st wave of the pandemic had passed, the prevalence of clinically relevant psychological distress decreased to 34.5%. In addition to sex and profession, the number of weekly hospital shifts with exposure to COVID-19 patients was predictive of clinically relevant psychological distress. COVID-19 exposure was low in July 2020, with a majority of participants reporting no workplace exposure in the previous week (Fig 1). In this context, ongoing exposure to COVID-19 was a significant predictor of mental health issues.

When data from T1, T2 and T3 were analysed longitudinally using a mixed effects model, the strongest predictor of clinically relevant psychological distress was pandemic timepoint. The inclusion of individual COVID-19 workload variables in the mixed effects model did not improve prediction. This suggests that the broader context of the pandemic had more impact on the psychological wellbeing of ICU HCW than did individual COVID-19 workload variables.

Factors predictive of persistent psychological distress were analysed by comparing participants who displayed psychological distress at both T1 and T3 with those who displayed

psychological distress at T1 but not at T3. The strongest predictor of persistent psychological distress was the number of weekly shifts with COVID-19 exposure. This is an important modifiable risk factor that could be addressed by providing modified scheduling for ICU HCW experiencing persistent psychological distress. It could also be addressed by rotating ICU HCW between hospitals with high and low patient numbers to limit the number of shifts with disease exposure.

RTs and allied health professionals were more likely to show recovery than physicians, while registered nurses were similar. Studies have shown that physicians are reluctant to seek help for psychiatric illness due to stigma and concerns about licensing [30]. In addition, ICU physicians in Canada often work alone, without immediate peer support. Social isolation is associated with mental health disorders, so a lack of close contact with peers may have impacted physician recovery [31].

Surprisingly, a higher median number of COVID-19 patients in the ICU was associated with recovery from psychological distress. Since patient loads were universally low at T3, this finding suggests that individuals who experienced psychological distress in the context of high patient loads at the beginning of the 1st wave were more likely to recover when patient loads decreased. Conversely, individuals who experienced psychological distress at T1 in the absence of high patient loads were less likely to recover.

One important variable that was not assessed in this study was pre-existing psychiatric disease. HCW suffer from anxiety and depression at rates that may exceed those of the general population [32]. Prior to the COVID-19 pandemic, symptoms of anxiety and depression were reported in 18% and 11–31% of American ICU nurses, respectively [33, 34]. Similarly, in France, symptoms of depression were reported in 18.8% of ICU physicians and 15.6% of ICU nurses and nursing assistants [35]. Although the GHQ-12 is designed to detect acute changes in mental health, it is possible that pre-existing mental illness contributed to the high rates of psychological distress detected in this study.

Only a few studies have monitored the mental health of ICU HCW longitudinally during the 1st wave of the pandemic. A study of 526 ICU HCW in the UK reported a decrease in clinically relevant psychological distress, from 50% to 34.6% of participants, between the acceleration and deceleration phases of the 1st wave of the pandemic [13]. Psychological distress was associated with concerns about being infected with COVID-19 and about infecting family members, however the investigators did not examine which variables were associated with persistence of psychological distress. A study of 786 frontline HCWs (not all of them from ICU) in the United States reported a similar decrease in psychological distress from 35.1% to 20.5% of participants between April and November 2020 [12]. Factors associated with persistent psychological distress were a history of psychiatric disease, fewer years of practice, the presence of post-surge stressors, and family-related concerns. Neither study evaluated the impact of COVID-19 workload on psychological distress.

Our study highlights the high rates of clinically relevant psychological distress amongst ICU HCW during the 1st wave of the pandemic, the importance of pandemic timepoint as a predictor of psychological distress, and the association between the number of shifts worked with COVID-19 exposure and persistent psychological distress.

## Strengths

This study has a number of strengths. Data was collected early in the 1st wave of the pandemic, prior to most published studies. Longitudinal data was also collected, allowing for mental health to be monitored over time. Participants came from a large number of centres, allowing for comparison between participants with variable COVID-19 workloads. Weekly data

collection for COVID-19 workload variables allowed for good data precision and minimized recall bias.

## Limitations

This study has several limitations. Participants were self-selected and may not be representative of Canadian ICU HCW; however, the study was publicized as a study of COVID-19 infection and transmission rather than mental health, which may have mitigated self-selection bias. The study population was relatively small with significant loss to follow-up; however, no differences with respect to demographics were noted across study timepoints. Moreover, the prevalence of psychological distress was similar amongst participants who completed all 3 GHQ-12 surveys, suggesting that loss to follow-up did not significantly impact results. No data is available with respect to pre-existing psychiatric illness. Although the GHQ-12 is designed to measure breaks in usual functioning, individuals with pre-existing mental health issues may have been more likely to report psychological distress.

## Conclusions

In this longitudinal study of Canadian ICU HCW, clinically relevant psychological distress decreased from 64.5% to 34.5% during the 1st wave of the COVID-19 pandemic. The strongest predictor of clinically relevant psychological distress was pandemic timepoint, suggesting that external factors, such as news media coverage, played a significant role in HCW mental health. The number of weekly shifts with exposure to COVID-19 was associated with persistent psychological distress at the end of the study period.

## Supporting information

**S1 Appendix. Study advertisement.**
(PDF)

**S1 Dataset. Study data in excel format.**
(XLSX)

**S1 Checklist. Checklist for Reporting Results of Internet E-Surveys (CHERRIES).**
(DOCX)

## Acknowledgments

We thank the Canadian ICU professionals who generously participated in this study. We thank Dr Marc de Rosnay and Dr Ana Marreiros for helpful discussions.

## Author Contributions

**Conceptualization:** Kyra Moura, Claire Moura, Frédérick D'Aragon, Jennifer L. Y. Tsang, Alexandra Binnie.

**Data curation:** Kyra Moura, Claire Moura.

**Formal analysis:** Daniel Pestana, Taylor Mouliakis, Jennifer L. Y. Tsang, Alexandra Binnie.

**Funding acquisition:** Jennifer L. Y. Tsang.

**Methodology:** Daniel Pestana, Frédérick D'Aragon, Jennifer L. Y. Tsang, Alexandra Binnie.

**Project administration:** Jennifer L. Y. Tsang, Alexandra Binnie.

**Writing – original draft:** Daniel Pestana, Kyra Moura, Claire Moura, Taylor Mouliakis, Frédérick D'Aragon, Jennifer L. Y. Tsang, Alexandra Binnie.

**Writing – review & editing:** Daniel Pestana, Kyra Moura, Claire Moura, Taylor Mouliakis, Frédérick D'Aragon, Jennifer L. Y. Tsang, Alexandra Binnie.

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
