## [Decision Letter · Decision Letter 0]

22 Jun 2023

PONE-D-23-09335The impact of COVID-19 workload on psychological distress amongst Canadian intensive care unit healthcare workers during the 1st wave of the COVID-19 pandemic: a longitudinal cohort study.PLOS ONE

Dear Dr. Tsang,

Thank you for submitting your manuscript to PLOS ONE. After careful consideration, we feel that it has merit but does not fully meet PLOS ONE’s publication criteria as it currently stands. Therefore, we invite you to submit a revised version of the manuscript that addresses the points raised during the review process.

We look forward to receiving your revised manuscript.

Kind regards,

Juan Jesús García-Iglesias, Ph.D.

Academic Editor

PLOS ONE

4. Thank you for stating the following in the Funding Section of your manuscript:

“This study was funded by an unrestricted research grant to JT from the non-profit Mohawk-Medbuy Corporation (https://www.mohawkmedbuy.ca/). The funders had no role in study design, data collection and analysis, decision to publish, or preparation of the manuscript.  “

Additional Editor Comments:

Please revise the manuscript according to the reviewers' comments and upload the revised file. Any revisions should be “clearly highlighted”, for example using the Track Changes function in Microsoft Word or using a different colour, so that they are easily visible to the editors and reviewers.

Please provide a short Cover letter detailing any changes, for the benefit of the editors and reviewers.

Reviewers' comments:

Reviewer's Responses to Questions

**Comments to the Author**

1. Is the manuscript technically sound, and do the data support the conclusions?

Reviewer #1: Yes

Reviewer #2: Yes

2. Has the statistical analysis been performed appropriately and rigorously? 

Reviewer #1: Yes

Reviewer #2: Yes

3. Have the authors made all data underlying the findings in their manuscript fully available?

Reviewer #1: Yes

Reviewer #2: Yes

4. Is the manuscript presented in an intelligible fashion and written in standard English?

Reviewer #1: Yes

Reviewer #2: Yes

5. Review Comments to the Author

Reviewer #1: In this paper, the authors present a longitudinal study of mental health among Canadian healthcare workers (HCWs) from intensive care units (ICU). The population was studied longitudinally at the beginning, middle and end of the first wave of Covid in 2020. A total 309 HCWs participate in the first survey timepoint, and around 190 at each of the two subsequent rounds of survey. Overall, the authors identified several factors that contributed to the presence of psychological distress at the various timepoints, as well as identifying factors associated with the likelihood of recovery from the distress.

The authors are to be commended for mounting such a comprehensive project at such short notice. The results are interesting, and the paper is generally well written.

I have some comments for the authors’ consideration:

Methods : page 4, lines 109-111 : were the data about workload self-reported? Or were they retried from written hospital administrative data? If self-reported, this may be subject to bias and should be mentioned as a limitation.

Methods / survey administration: I appreciate that the authors provided additional methodological information in the supplementary material, but to facilitate the reading of the main paper, more detail is needed here, on two key points:

First, the authors should explain exactly how the survey was distributed – was it simply emailed to all registered members of the societies listed? Do the authors have any idea of the number of people who received the link? Could recipients forward the link to other people?

Second, it is unclear what exactly the participants were required to do. From reading the results on page 6, paragraph “Covid 19 workload”, and from my reading of the consent form provided in the supplementary material, it appears that the participants were asked to provide information about workload on a weekly basis. This is not clear from the methods, where the reader is given to understand that the participants were surveyed at 3 timepoints only (and not in between). The authors should clearly explain what was measured at the 3 timepoints T1, T2 and T3, and specify whether the participants also had to provide information about their workload every week. If so, please explain how this was done. It’s not explained in the supplementary methods either. I cannot understand fully, from the methods presented, whether the participants were individually identified and followed – i.e. are the 195 and 191 respondents who participated in the surveys at T2 and T3 respectively, a subset of the original 309 participants from T1? If they were all recording their workload weekly, that indicates a high drop-out rate, with about one third of the participants lost by T2.

Results, page 8, line 231/232: if weekly hospital shifts is associated with a protective effect, then it should be specified in Table 3 that the OR is the increase in risk per additional shift worked. Same remark for weeks shifts with Covid exposure.

Results, page 10, line 248 – do the authors mean that there are 190 unique individuals who were identified to have answered both surveys?

Table 4 – I don’t recall it being mentioned in the methods that the BRCS was administered – when was this tool administered to participants? This should be added to the methods section.

Table 5 – I think the title of this table is wrong, or else the authors have interpreted the data the wrong way around (unlikely). On page 11, lines 272-274, the authors report the factors associated with the likelihood of recovery from psychological distress, and these factors are what Table 5 purports to report. However, the authors report that male sex and a higher number of Covid patients in the ICU were associated with recovery – but this is ambiguous. According to the odds ratios reported in Table 5, both these factors have ORs that are associated with a protective effect, and therefore, a lower likelihood of the outcome. If the outcome modelled in the regression analysis is “recovery”, then male sex and more COVID cases in ICU “protect against” recovery and thus, are associated with a lower likelihood of recovery. This is incongruent with the result report on lines 273/274, where the authors state that physician profession and a higher number of weekly shifts with exposure to Covid patients were associated with persistence of psychological distress. Firstly, there is no odds ratio for the profession “physician”, indicating that this category was the reference category, so it cannot be associated with the outcome. There must a mistake somewhere. Secondly, the odds ratio associated with the number of hospital shifts with exposure is 1.97, indicating an approximately twofold increase in the risk of the outcome. This suggests that the outcome being modelled in the table is “persistent psychological distress” (i.e. lack of recovery), which is the opposite of what the table title indicates. So, overall, there is some work to be done here to ensure that the table title clearly indicates the outcome being modelled, to enable the reader to interpret the ORs in the right direction. Second, the authors must clarify the reference category used among the professions, and correct the result pertaining to physicians. The same comment applies to the discussion, because the authors discuss this result about physician recovery (page 12, line 335), but the reader does not see this result in the results section.

A key limitation of this study is that there is no information about baseline mental health among the respondents. Indeed, it has been reported that pre-pandemic, around 13% of physicians/surgeons and around 18% of registered nurses may have had diagnosed depression (DOI: 10.1097/JOM.0000000000002630). Therefore, we do not know how many of these professionals had mental health issues or signs and symptoms of burnout before the pandemic began.

Discussion, page 12, line 323: the authors state that “these participants appear to have been more susceptible to mental health issues at T3” – what is your basis for this assertion?

Limitations – as mentioned above, the authors might add the lack of knowledge about baseline mental health status, and possible bias in the reporting of workload. Also, since we are now more than three years later, the authors might mention how this information can be of use to the medical community now.

References: the authors should verify the format of the references – some still contain dates (e.g.refs 5,6, 7….), and some journal names are not presented correctly (e.g. ref 10…)

In summary: The main action points include a revision of the methods to make the exact procedures clearer; clarify the interpretation of Table 5 and the related text in the results and discussion; and add limitations as detailed above.

Reviewer #2: Thank you for the opportunity to review this manuscript. I commend the authors on conducting this important study and at such an early and chaotic phase of the pandemic. It is impressive that they had a relatively high follow-up rate as part of this longitudinal survey, which is notoriously challenging to achieve, especially in studies of healthcare providers.

I have included suggestions below for the authors to consider.

Abstract

- The abstract does not currently include statistical data from the multivariable analysis, although the findings are described in writing. Generally, the data are included for readers to take the strengths of any associations into consideration. Would suggest the authors include the few relevant data points numerically for findings that they have summarized in writing.

Introduction

- The Introduction is well-written and referenced well. Given that this is a Canadian study with a large proportion of participants from Ontario, perhaps the authors can consider summarizing other Canadian/ Ontario studies that have been published in the literature... numerous have been published that can be incorporated... a few of these listed below:

> PMID: 34940952

> PMID: 35167590

> PMID: 36627462

- The last paragraph of the Introduction section begins with "Only a few studies..." but these studies have not been cited. Please cite the studies you're referring to.

- The objective statement (last 2 sentences in the introduction) are detailed enough that they are similar to statements in the Methods. I would suggest a broad objective statement be included instead to summarize the overall purpose of the study, without getting into Methodology at this stage.

Materials and Methods

- The study is well described in this section.

Results

- What proportion of participants were from academic vs. community settings? This is important to note given my comment below regarding the authors' speculation about community ICU physicians.

- The authors describe the findings of the BRCS in Table 4 but did not indicate in the Methods that this instrument was also administered and do not discuss its findings in the Results. This is an interesting variable and would be of interest to readers to know about the outcomes on the resilience front. I would suggest discussing this instrument in the Methods and Results in more detail.

- Also regarding the BRCS, was this variable considered for inclusion in the MV analysis?

- Table 3 - I would suggest putting the most notable findings where a variable was predictive of the outcome in bold font for easier review by readers.

- Table 5 indicates that the physician profession was used as the reference variable for the analysis but the text states that the physician profession was predictive of persistence - can the authors please explain and provide the data that led to this conclusion? Was there another analysis in which the physician profession was not used as the reference variable and allowed the authors to draw this conclusion?

- The authors state that physician role was predictive of persistence of mental health issues - was the nursing profession (which showed the highest proportion of respondence with persistence) not predictive of persistence? If not, how do the authors explain this finding? Is it possible that this finding could be explained by multicollinearity between sex and the nursing profession (since most nurses were female)? I imagine that would influence the MV analysis and perhaps, it may not be appropriate to include both nursing profession and sex in the same model due to high collinearity issues?

Discussion

- As noted in my comment about the Introduction, it would be great if the authors could discuss their findings in the context of other Canadian-based studies.

- The first paragraph of the Discussion section is a reiteration of the Intro and Methods - I would propose this is not necessary and the Discussion should usually begin with a high-level summary of the study findings.

- The authors discuss the isolated work environment of community ICU physicians but did not state whether this variable specifically (being a community ICU physician) was an independent predictor of poorer mental health. Perhaps the MD group has a small sample size and would not allow for such an analysis. At a minimum, would suggest reporting the number of community vs. academic physicians in the sample.

Conclusion

- Here, the authors discuss 'external factors, such as news media coverage...' which they don't explicitly describe in the Discussion section as potential contributors to poorer mental health. I would suggest the authors discuss this in more detail in the discussion, so that they are not discussing a new concept in the Conclusion section.

6. PLOS authors have the option to publish the peer review history of their article (what does this mean?). If published, this will include your full peer review and any attached files.

Reviewer #1: **Yes: **Fiona Ecarnot

Reviewer #2: No

---

## [Author Response · Author response to Decision Letter 0]

8 Aug 2023

Thank you very much for allowing us to resubmit our manuscript. Please see our detailed to Response to Reviewers in the documents list.

---

## [Editor Report · Decision Letter 1]

15 Aug 2023

The impact of COVID-19 workload on psychological distress amongst Canadian intensive care unit healthcare workers during the 1st wave of the COVID-19 pandemic: a longitudinal cohort study.

PONE-D-23-09335R1

Dear Dr. Tsang,

We’re pleased to inform you that your manuscript has been judged scientifically suitable for publication and will be formally accepted for publication once it meets all outstanding technical requirements.

Kind regards,

Juan Jesús García-Iglesias, Ph.D.

Academic Editor

PLOS ONE
---

## [Editor Report · Acceptance letter]

27 Feb 2024

PONE-D-23-09335R1 

PLOS ONE

Dear Dr. Tsang, 

I'm pleased to inform you that your manuscript has been deemed suitable for publication in PLOS ONE. Congratulations! Your manuscript is now being handed over to our production team.

Kind regards, 

on behalf of

Dr. Juan Jesús García-Iglesias 

Academic Editor

PLOS ONE